# Whole-Transcriptome Profiling and Functional Prediction of Long Non-Coding RNAs Associated with Cold Tolerance in Japonica Rice Varieties

**DOI:** 10.3390/ijms25042310

**Published:** 2024-02-15

**Authors:** Hao Wang, Yan Jia, Xu Bai, Weibin Gong, Ge Liu, Haixing Wang, Junying Xin, Yulong Wu, Hongliang Zheng, Hualong Liu, Jingguo Wang, Detang Zou, Hongwei Zhao

**Affiliations:** Key Laboratory of Germplasm Enhancement and Physiology & Ecology of Food Crop in Cold Region, Ministry of Education/College of Agriculture, Northeast Agricultural University, Harbin 150030, China; 18946076063@163.com (H.W.); baixu2021@163.com (X.B.); g314159261201@163.com (W.G.); lgneau@163.com (G.L.); wanghaixin0504@163.com (H.W.); m18232228031_1@163.com (J.X.); 18800438267@163.com (Y.W.); zhenghongliang008@163.com (H.Z.); liuhualongneau@163.com (H.L.); 55190292@163.com (J.W.); zoudtneau@126.com (D.Z.)

**Keywords:** rice, temperature, LncRNA, MiRNA, competing endogenous RNA

## Abstract

Low-temperature chilling is a major abiotic stress leading to reduced rice yield and is a significant environmental threat to food security. Low-temperature chilling studies have focused on physiological changes or coding genes. However, the competitive endogenous RNA mechanism in rice at low temperatures has not been reported. Therefore, in this study, antioxidant physiological indices were combined with whole-transcriptome data through weighted correlation network analysis, which found that the gene modules had the highest correlation with the key antioxidant enzymes superoxide dismutase and peroxidase. The hub genes of the superoxide dismutase-related module included the UDP-glucosyltransferase family protein, sesquiterpene synthase and indole-3-glycerophosphatase gene. The hub genes of the peroxidase-related module included the WRKY transcription factor, abscisic acid signal transduction pathway-related gene plasma membrane hydrogen-ATPase and receptor-like kinase. Therefore, we selected the modular hub genes and significantly enriched the metabolic pathway genes to construct the key competitive endogenous RNA networks, resulting in three competitive endogenous RNA networks of seven long non-coding RNAs regulating three co-expressed messenger RNAs via four microRNAs. Finally, the negative regulatory function of the WRKY transcription factor *OsWRKY61* was determined via subcellular localization and validation of the physiological indices in the mutant.

## 1. Introduction

Rice (*Oryza sativa* L.) is an essential cereal crop worldwide [1]. However, low-temperature stress is the main abiotic factor restricting rice production [2,3].

Particularly in middle- and high-latitude areas, chilling injury occurs frequently. Additionally, low-temperature damage in the nutritional growth stage of rice can delay all the fertility stages and tasseling [4]. Low-temperature cold damage in the reproductive growth stage can also cause pollen sterility and a significant decrease in the fruit set [5,6]. Therefore, due to cold damage, China’s estimated annual rice production is reduced by 3–5 billion kg [7]. Cold tolerance also differs between the nutritional and reproductive growth periods of rice, which may have regulatory mechanisms in opposite directions. During the early growth stage of rice, low-temperature cold damage inhibits seedling establishment, leading to scheduled crop maturation failure. This results in significant yield reductions or early snowfall resulting in yield extinction. Previously, studies on regulating cold tolerance genes mainly focused on protein-coding genes. For instance, *OsbHLH002/OsICE1*, a transcription factor (TF) highly homologous to *ICE1* in *Arabidopsis*, regulates cold tolerance in rice by targeting *OsTPP1* to control trehalose biosynthesis [8]. *OsCNGC9* is a cyclic nucleotide-gated channel, and the loss-of-function *OsCNGC9* mutant was sensitive to prolonged cold treatment and defective in cold-induced calcium inward flow. At the same time, its overexpression enhanced the cold tolerance of the cells [9]. However, there are few studies on the function of non-coding RNA in rice under cold stress.

Under adverse conditions, such as low temperature, the intracellular oxygen metabolism of rice is imbalanced and reactive oxygen species (ROS), including superoxide anion O^2−^, hydrogen peroxide H_2_O_2_, and hydroxide ion hydroxyl radical (OH^−^), are produced. This process triggers membrane lipid peroxidation, damaging the cell membrane system [10]. ROS also promote polyunsaturated fatty acid degradation and malondialdehyde (MDA) production, further damaging plant tissues and cells [11]. Two major systems protect rice against oxidative stress: enzymatic and non-enzymatic systems. The enzymatic system includes various antioxidant enzymes that catalyze ROS-scavenging reactions. They include superoxide dismutase (SOD), peroxidase (POD), and catalase (CAT), some of the most effective antioxidant enzymes. These enzymes can turn superoxide anions and hydrogen peroxide into water and oxygen molecules, thus mitigating ROS damage to cells [12]. Non-enzymatic systems include various antioxidants, among which reduced glutathione and ascorbic acid are the most important [13,14]. Other non-enzymatic systems include soluble sugars, soluble proteins, and proline. Soluble sugars can act as osmoregulatory substances for cells in low-temperature adversity, stabilize cell membranes and protoplasmic colloids, and provide carbon skeletons and energy to synthesize other organic substances [15,16]. Overexpression of *OsTPP1*, *OsTPP2*, and *OsTPS1* (key genes for algal sugar synthesis) significantly improved low-temperature tolerance in rice [17,18]. Rice also accumulates substantial amounts of proline at low temperatures. Proline is widely involved in osmoregulation, carbon and nitrogen metabolism, and protecting most enzymes from denaturation and inactivation [19]. Also, proline is crucial to stabilizing polyribosomes and maintaining protein synthesis [19]. Overexpression in plants of the *OsCOIN*, *OsMYB2*, *OsMYB4*, *OsMYB3R-2*, and *OsZFP245* genes significantly increased the proline content and enhanced tolerance to low temperatures [20,21,22,23].

Long non-coding RNAs (lncRNAs) form a class of RNAs with a length of more than 200 nucleotides; they cannot encode proteins [24,25]. Based on the location of the coding genes and lncRNAs in the genome, lncRNAs can be classified into four types: (1) antisense lncRNAs; (2) intergenic lncRNAs; (3) intronic lncRNAs; and (4) sense lncRNAs [26]. According to the mechanism of action, lncRNAs can be broadly classified into three categories: transcriptional regulation, post-transcriptional regulation, and other functions [27]. In plants, most identified lncRNAs have similar characteristics to messenger RNAs (mRNAs); they are capped at the 5’ end, polyadenylated at the 3’ end, and produced mainly by RNA polymerase II [28]. However, lncRNAs also have some minor differences from mRNAs in many aspects. For example, lncRNAs are shorter and have fewer exons than mRNAs transcripts. They also have lower expression levels, higher natural expression variation, and lower evolutionary conservation among species; there is no interspecies sequence conservation [29,30,31,32].

As important regulators of gene expression, lncRNAs play important roles in plant development and the response to adversity, including flowering time [33], nutrient metabolism [34], sexual reproduction [35], abiotic stress response [36], and pathogen resistance. The biological functions of some plant lncRNAs have been studied and validated. For example, *COOLAIR*, a natural antisense transcript, and *COLDAIR*, an intronic lncRNA, were reported to repress the expression of *FLOWERING LOCUS C* (FLC) through histone modification; this affects vernalization in *Arabidopsis* [37]. Instead of acting as a transcriptional regulator, lncRNA *IPS1* regulates phosphate uptake by binding to *miR399* to reduce *miR399*-mediated cleavage of phosphate 2 [38]. In hybrid rice, long-day-specific male-fertility-associated RNA could regulate photoperiod-sensitive male sterility. The decreased long-day-specific male-fertility-associated RNA expression caused early programmed cell death (PCD) in developing anthers under long-day conditions and then resulted in photoperiod-sensitive male sterility [39]. *MuLnc1* was cleaved by mul-*miR3954* to generate *si16157*9, which was a ta-siRNA (trans-acting siRNA), and could regulate plant biotic and abiotic resistance by cleaving the transcripts of calmodulin-like protein gene *CML27* [36]. LncRNAs may respond to *Dickeya zeae* infection in rice by regulating the transcript levels of their targets and microRNAs (miRNAs) [25]. In winter wheat varieties, *lncRNAR9A* was predicted to interact with *miR398*; *miR39*8 and *lncR9A* were transferred into *Arabidopsis thaliana*. The *lncR9A*-transferred plants were found to have lower MDA content and higher SOD content than the wild type, and to have significantly enhanced cold resistance. These results suggest that lncRNA indirectly regulates the expression of *CSD1* through competitive binding of *miR398* to affect plant cold resistance. Scientists have conducted numerous studies on the potential genetic mechanisms of resistance to low-temperature stress in rice. However, there are few reports on the involvement of lncRNAs in the regulation of low temperature in rice. No competitive endogenous RNA (ceRNA) network has been reported for rice in cold tolerance, and it is unclear whether the lncRNA–miRNA–mRNA target interaction network is involved in the response of rice to low-temperature stress.

In this study, we aimed to establish the ceRNA network of rice’s low-temperature response and elucidate the transcriptional regulatory network and physiological mechanism of rice leaves in response to low-temperature stress. Therefore, we sequenced the entire transcriptome RNA and determined the antioxidant system indicators under low temperatures for Jilin Sunset (JL, a cold-tolerant japonica rice variety) and Jinhe (JH, a low cold-sensitive japonica rice variety) (Figure 1A). The key module hub gene and significant enrichment pathway genes were screened via weighted correlation network analysis (WGCNA) to construct ceRNA networks, and seven lncRNAs were obtained to regulate three co-expressed mRNAs through four miRNAs. In addition, the function of the hub gene WRKY TF *OsWRKY61* (*Os11g0685700*) was verified by means of subcellular localization and mutational physiological index assays. This study’s results lay the foundation for studying non-coding RNAs in japonica rice and provide insights into the mechanism of cold-tolerance regulation in japonica rice.

## 2. Results

### 2.1. Effects of Low-Temperature Stress on Physiological Indexes of Rice

#### 2.1.1. Effect of Low-Temperature Stress on Morphology of Rice

Under normal growth conditions, the leaves of JH and JL spread out and the degree of leaf curling (DLC) was 0. The difference in plant height between JH and JL was 0.87 cm, and there was no obvious difference in plant morphology. After 12 h of low-temperature stress, the leaves of JH were significantly crimped, but there was no significant change in JL. The DLC of JH was 53.62%, and the DLC of JL was 9.57%. The difference in plant height between JH and JL was 2.13 cm. After 24 h of low-temperature stress, the DLC of JH was 81.31% and that of JL was 13.63%. The difference in plant height between JH and JL was 2.65 cm. After 48 h of low-temperature stress, the difference in plant morphology was the greatest, the DLC of JH was 87.01%, the DLC of JL was 15.63%. The difference in plant height between JH and JL was 2.95 cm (Figure 1A).

#### 2.1.2. Effect of Low-Temperature Stress on Osmoregulatory Substances in Rice

As illustrated in Figure 1B, the proline, soluble sugar (SS), and soluble protein (SP) content accumulated with the cold treatment time and peaked at 48 h. The proline accumulation rate and content in JL were significantly higher than in JH. Compared with the control, after 48 h of cold treatment, the proline, SS and SP content in JH was 4.69, 1.74, and 2.08 times higher than that at the stress initiation, respectively. At this time, JL was 6.40, 2.66, and 3.06 times higher than at the stress initiation. This indicates that the cold-tolerant variety, JL, can rapidly respond to low-temperature stress and maintain a relatively high osmoregulatory substance content (Figure 1B).

#### 2.1.3. Effect of Low-Temperature Stress on Antioxidant Enzymes in Rice

The CAT, POD and total antioxidant enzyme (T-AOC) of both species increased continuously with the cold treatment time and peaked at 48 h. The CAT, POD and T-AOC accumulation rate and content in JL were significantly higher than in JH. Compared with the control, after 48 h of cold treatment, the CAT, POD and T-AOC content in JH was 1.44, 1.60, and 3.47 times higher than that at the stress initiation, respectively. At this time, JL was 2.00, 2.16, and 4.94 times higher than at the stress initiation. These results indicate that the cold-tolerant variety, JL, can accumulate CAT and POD rapidly in response to low-temperature stress and maintain relatively high levels to remove H_2_O_2_ from leaf tissues and inhibit peroxide damage to plants (Figure 1B).

During 0–24 h of cold stress, the SOD contents of both varieties increased significantly, with a greater increase in the JL. At 24 h, the SOD contents in the cold-sensitive variety, JH, was 2.87 times higher than at the beginning of the stress, while this in the cold-tolerant variety, JL, was 3.24 times higher than at the beginning of the stress. Furthermore, at 24–48 h, the SOD contents in JL increased and remained at high levels. Nonetheless, the SOD times in JH decreased sharply with prolonged cold damage. At 48 h, the SOD in JH was 2.65 times higher than at the beginning of the stress and 4.34 times higher than at the beginning of the stress in JL. This indicates that the cold-tolerant variety, JL, can remove excessive reactive oxygen radicals and enhance the cold resistance of plants by maintaining high SOD levels (Figure 1B).

#### 2.1.4. Effects of Low-Temperature Stress on Oxidative Stress of Rice

During cold stress, the OH^−^and MDA contents in both species increased continuously, and the accumulation rates and contents of OH^−^and MDA in JH were significantly higher than those in JL. Compared with the control, the OH^−^and MDA contents in JH were 1.84 and 4.40 times higher than at the beginning of the stress after 48 h of cold treatment. In JL, they were 1.42 and 3.36 times higher than at the beginning of the stress, respectively, indicating that the cold-tolerant variety, JL, could scavenge OH^−^and MDA during the-low temperature stress; thus, maintaining OH^−^and MDA at a low level (Figure 1B).

#### 2.1.5. Identification of Important Antioxidant Phenotypes

The variable importance in projection (VIP) values are orthogonal partial least squares discriminant analysis variable projection importance values that can measure the contribution of individual phenotypic changes to the classification, with larger VIP values contributing more. Usually, a VIP value > 1 is a screening criterion [40]. Orthogonal partial least squares discriminant analysis was performed at different time points of cold treatment based on the nine major metabolites in response to low temperature, according to the VIP scores. The VIP scores of the POD, SOD and OH^−^were >1, and they were identified as the most essential antioxidant phenotypes among them (Figure 1C, D).

### 2.2. High Throughput Sequencing

To systematically identify the expression profiles of the lncRNAs and mRNAs associated with cold tolerance in Japonica rice seedlings, we performed strand-specific whole-transcriptome RNA-seq on control and cold-treated rice seedling leaves of two Japonica varieties using Illumina machines. All the samples (2 varieties × 5 time points × 3 replicates) were analyzed in three independent biological replicates. After using Trimmomatic software (v0.32) to remove any low-quality and contaminated readings, all the samples from JH and JL yielded nearly 3.3 billion high-quality reads (approximately 100 million per sample on average). The mapping of clean reads in each library to the Oryza sativa reference genome was 95.97–97.86%. In addition, all the samples had Q20 values >97.8%, Q30 values >93.64%, and GC content of 46.55–48.21% (Appendix A), indicating the reliability of the RNA-seq data. These results provide a wealth of data for further analysis of the expression profiles and metabolic pathways of the mRNAs and lncRNAs.

### 2.3. Comparative Analysis of lncRNAs and mRNAs

The identified mRNAs and lncRNAs are widely distributed on all the chromosomes, among which the number of lncRNAs and mRNAs on chromosome 1 is the largest. In contrast, the lncRNAs and mRNAs on chromosomes 9, 10, and 12 are relatively few (Figure 2A). In addition, we further analyzed the basic characteristics of the identified lncRNAs and mRNAs in rice, as presented in Figure 2B. The lengths of the lncRNAs were 200–200,538 nt, with an average of 1362 nt. Most lncRNAs were <2000 nt in length, with the most lncRNAs <1000 nt in length, in decreasing order. On the other hand, the mRNA lengths were 30–57,648 nt, with an average of 3038 nt, and their lengths were mainly concentrated within 3000 nt. Most lncRNAs had only one exon, while the proportion of mRNAs with over two exons was significantly higher than that of the lncRNAs (Figure 2C). These results suggest that there are some differences in the characteristics of the mRNA and lncRNA in rice leaves.

### 2.4. Identification of Differentially Expressed mRNAs

The inter-sample agreement was examined using the Pearson correlation coefficient (R-value) between both varieties. The minimum R-value for each variety was 91.49%, and the maximum was 99.31% for three biological replicates of each variety compared. Replicate samples revealed close correlations, proving the reproducibility of the samples and their continued use for subsequent analysis. We compared the clean reads of each sample with the reference genome and obtained 36,850 mRNAs. According to the screening criteria of a fold change ≥ 2 and FDR ≤ 0.01, 12,817 and 15,806 mRNAs were identified from JH and JL, respectively, compared with the control 0 h, among which 8088 genes in JH were significantly up-regulated and 4729 were significantly down-regulated. Additionally, 10,446 genes in JL were significantly up-regulated and 5363 were significantly down-regulated. The number of up-regulated differential mRNAs was much larger than that of the down-regulated differential mRNAs. With prolonged cold treatment time, the number of differential mRNAs gradually increased, indicating the changed gene expression of rice seedlings with prolonged cold treatment time (Figure 3A).

We also compared both varieties at the same time point. We observed that 3102 genes (including 70 TF-encoding genes) were highly expressed, and 3028 genes (including 82 TF-encoding genes) displayed significantly low expression than JH. Similarly, the number of differentially expressed genes in the varieties at 48 h was the highest (Figure 3B). Some TF families displayed significant differences in the number of significantly highly and lowly expressed members. For instance, the expression of TF family members involved in abiotic stresses in plants, such as MYB, HB, and NAC, was significantly elevated. On the other hand, TFs with lower expression were mainly from the WRKY, bZIP, and MADS families (Figure 3C). The expression profiles of these TF families are presented in Appendix A. Finally, KEGG enrichment analysis was performed on all the DEmRNAs in the two varieties, and it was found that the main enrichment was in the biosynthesis of secondary metabolites, metabolic pathways, plant-pathogen interaction, alpha-linolenic acid metabolism, fatty acid elongation, photosynthesis-antenna proteins, plant hormone signal transduction, amino sugar and nucleotide sugar metabolism, glycosylphosphatidylinositol (GPI)-anchor biosynthesis and phenylpropanoid biosynthesis (Appendix A).

### 2.5. Identification of Differentially Expressed lncRNA

To characterize the putative lncRNAs of rice, first, the transcripts with length <200 nt were filtered. Second, the coding potential of the remaining transcripts was evaluated using Rfam, Pfam, and CPC2, and the intersection of non-protein-coding transcripts predicted using the three software were the candidate lncRNAs for further analysis. Finally, 7087 lncRNAs were identified in JH and JL. Based on the position of the lncRNAs in relation to the protein-coding genes, we classified them into four types: (3035, 42.82%) antisense lncRNA; (2700, 38.1%) intergenic lncRNAs; (548, 7.74%) intronic lncRNAs; and (804. 11.36%) sense lncRNAs (Figure 4A).

For each transcribed region, transcribed fragments per kilobase per million reads (FPKM) values can be estimated using StringTie software (version 2.2.0) to quantify its expression abundance and variation. The FPKM distribution of the 10 treatments revealed their basic expression patterns. As illustrated in Appendix A, the density–FPKM curves revealed the same trend in all the samples, indicating that the lncRNA expression was comparable among the groups.

According to the screening criteria of a fold change ≥ 2 and FDR ≤ 0.01, we first identified differential genes at each cold treatment time point of each variety. Compared with the control 0 h, 1306 and 2414 differential lncRNAs were identified in JH and JL, respectively. Additionally, 75.1% of the genes in JH were significantly up-regulated and 24.9% were significantly down-regulated. In JL, 59.4% of the genes were significantly up-regulated and 40.6% were significantly down-regulated (Figure 4B). In comparisons, the total number of significant regulatory genes differed. However, the number of up-regulated genes was higher than that of the down-regulated genes in all the groups. The number of differential lncRNAs in the late cold-treatment period was significantly higher than that in the early period, suggesting to some extent that these differentially expressed lncRNAs (DElncRNAs) may be involved in the response of rice to low temperature.

At the same time, we also compared both varieties at the same time point. Compared with JH, 726 genes were significantly highly expressed and 951 genes had significantly lower expression. The differential JL and JH genes were the highest at 4 h and lowest at 12 h (Figure 4B).

Targeted analysis showed that 4960 DEmRNAs were cis-regulated by these DElncRNAs, and 5803 DEmRNAs were trans-regulated by these DElncRNAs. These target genes are mainly enriched in the photosynthesis-antenna proteins, alpha-linolenic acid metabolism, plant-pathogen interaction, metabolic pathways and inositol phosphate metabolism (Appendix A).

### 2.6. Identification of Differentially Expressed miRNAs

We also performed small RNA sequencing on all the samples and obtained 409,055,593 clean reads. As a result, 542 existing miRNAs were identified using Bowtie (version 1.1.2) software, of which 409 were known miRNAs and 133 were novel miRNAs (Appendix A). Most of them were 21 nt (49.82%) and 22 nt (20.30%) in length (Figure 5A). Using edgeR software v3.24.3, with a fold change ≥ 1.5 and *p*-value ≤ 0.05 as the screening criterion, there were (41, 32, 55, 24), (33, 34, 27, 36) differential miRNAs in JH and JL at 0 h vs. 4 h, 0 h vs. 12 h, 0 h vs. 24 h, and 0 h vs. 48 h, respectively. At the same time, we also compared both varieties at the same time point. Compared with JH, 186 genes were significantly highly expressed and 157 genes had significantly lower expression (Figure 5B).

### 2.7. Identification of Weighted Gene Co-Expression Network Analysis Modules Associated with Cold Tolerance

To comprehensively analyze the relationship between the differential genes and cold tolerance phenotypes, we performed a weighted WGCNA. A gene-clustering scheme with a power of seven was constructed. Genes with similar expression patterns after filtering were divided into 17 modules (Figure 6A). Each module’s gene expression profile was correlated with all the samples to generate the heat map of the mode sample matrix (Figure 6B). After combining, the antioxidant physiological indices were combined with whole-transcriptome data through weighted correlation network analysis, and we found that the gene modules had the highest correlation with the key antioxidant enzymes SOD (r = −0.90, *p* = 4 × 10^−4^) and POD (r = −0.85, *p* = 0.005). SOD and POD were antioxidant phenotypes with VIP scores > 1 (Figure 1C,D), further proving the reliability of selecting both modules.

### 2.8. Identification and Analysis of Key Genes for Cold Tolerance

There were 1130 expressed genes in the SOD-related module and 2253 expressed genes in POD-related module. According to TopGO enrichment analysis, the genes in the key co-expression modules are significantly enriched in numerous pathways that can be classified into three major categories: biological processes, cellular components, and molecular functions.

The genes in the SOD-related module were significantly enriched in terpene synthase activity, oxidoreductase activity, acting on the CH-NH group of donors, oxygen as acceptor, abscisic acid binding, protein phosphatase inhibitor activity, glutathione transferase activity, magnesium ion binding, anthocyanin-containing compound biosynthetic process, regulation of protein serine/threonine phosphatase activity, glutathione metabolic process, and toxin catabolic process (Appendix A). The genes in the POD-related module were significantly enriched in ADP binding, polynucleotide adenylyltransferase activity, RNA-directed DNA polymerase activity, iron ion binding, mitochondrion, plastid, DNA integration, RNA-dependent DNA replication, apoptotic process, and mRNA polyadenylation (Appendix A). We also performed KEGG enrichment analysis, selecting the top 10 pathways in terms of the *p*-value. The genes in the SOD-related module were mainly concentrated in flavonoid biosynthesis, galactose metabolism, stilbenoid, diarylheptanoid, gingerol biosynthesis, endocytosis, arginine, and proline metabolism, phosphatidylinositol signaling system, beta-alanine metabolism, other types of O-glycan biosynthesis, selenocompound metabolism, and phenylalanine metabolism. The genes in the POD-related module were mainly concentrated in the linoleic acid metabolism, valine, leucine, and isoleucine biosynthesis, homologous recombination, pyrimidine metabolism, purine metabolism, pantothenate and CoA biosynthesis, pyruvate metabolism, 2-Oxocarboxylic acid metabolism, nucleotide excision repair, and mismatch repair (Figure 7A,B).

To identify the hub genes controlling cold tolerance in rice, linkage count analysis was performed on genes within the key module, and genes with high linkage counts were considered very important. The top five genes with the highest connectivity in each module were selected as hub genes. The hub genes in the SOD-related module included a UDP-glucosyltransferase family protein (*Os05g0527000*), a sesquiterpene synthase (*Os08g0167800*), an indole acetic acid (IAA)-related indole-3-glycerophosphatase gene (*Os03g0797500*), a gene encoding oxidoreductase (*Os04g0339400*) and an unknown gene (Os05g0212900). The hub genes in the POD-related module included a WRKY TF (*Os11g0685700*), a plasma membrane hydrogen-ATPase (*Os02g0825600*) related to the abscisic acid (ABA) signaling pathway, a receptor-like kinase (*Os02g0615800*), and two unknown genes, *Os03g0103950* and *Os08g0288050*.

### 2.9. Construction and Analysis of lncRNA-miRNA-mRNA ceRNA Network

Using TargetFinder (Version 2.0) to compare the sequences of all the types of RNAs obtained via sequencing on 1 July 2022, we observed relationships between 1138 lncRNAs, 5601 mRNAs, and 225 miRNAs (Appendix A). To accurately analyze the role of these RNAs during the cold rice response, we intersected the fraction of differences in these relationship pairs with genes in the significantly related module in WGCNA. In total, 85 intersecting differentially expressed mRNAs (DEmRNAs) were obtained in the SOD-related module (Appendix A) and 461 intersecting DEmRNAs were obtained in the POD-related module (Appendix A). In the DEmRNAs intersection of the SOD- and POD-related modules, we selected some hub genes of the module screened above and the differential genes mainly related to cold tolerance in the enrichment pathway. The network consisted of three mRNAs, four miRNAs, and seven lncRNAs. Finally, we constructed the ceRNA network using Cytoscape3.3.0 software (Figure 8).

### 2.10. Validation of Differentially Expressed Genes in RNA-Seq Using qRT-PCR

We compared the log_2_ fold changes of six selected differentially expressed genes (DEGs) between RNA-Seq and real-time quantitative polymerase chain reaction (qRT-PCR) to validate the RNA-Seq results, demonstrating a correlation coefficient (R^2^) of 0.8881 (Figure 9). The genes and primer sequences used for the PCR reactions are listed in (Appendix A). The qRT-PCR data were consistent with the RNA-Seq data, indicating the reliability of the RNA-Seq results.

### 2.11. OsWRKY61 Is Localized in the Nucleus and Chloroplasts

*OsWRKY61* is not only a key candidate mRNA in this study but also a key mRNA in the ceRNA network, and the expression of this gene is up-regulated in the cold-sensitive variety, JH, and down-regulated in the cold-tolerant variety, JL. Therefore, clarifying the role of *OsWRKY61* is important for determining mRNA function as well as the prediction of lncRNA and miRNA.

To determine the subcellular localization of *OsWRKY61*, we expressed the GFP-*OsWRKY61* fusion construct in *Arabidopsis* protoplasts and found that the control vector (35S: GFP) had a bright GFP signal green distributed throughout the whole cell. However, the fluorescence of 35S: GFP: *OsWRKY61* fusion protein was only localized in the nucleus and chloroplasts (Figure 10), indicating that *OsWRKY61* may be localized in the nucleus and chloroplasts.

### 2.12. Mutant Construction and Phenotype Analysis

To investigate the effect of *OsWRKY61* knockdown on rice growth, we used the CRISPR-Cas9 method to construct *OsWRKY61* mutants. A CRISPR/Cas9 vector targeting the *OsWRKY61* coding region guide RNA was first constructed and transformed into rice (ZhongHua 11) (Figure 11). Then, PCR amplification and sequencing were performed for each generation of regenerated knockout plant leaves, and two mutation types were screened for T3 generation pure plants with 1 bp insertion and 1 bp deletion, respectively (Figure 11A,B), which resulted in the early termination of code shifting and translation. We also used online tools (http://skl.scau.edu.cn/offtarget/, accessed on 2 January 2024) in the rice genome search and *OsWRKY61* to target highly similar sequences to test for potential off-target cutting on 7 August 2022, with miss cut not detected.

POD, proline, and MDA are the key indicators of the physiological state of cells under cold stress. Compared with the control (room temperature treatment), the POD, PRO, and MDA contents were increased after 48 h of cold treatment, but the knockout strain had significantly higher POD and proline and significantly lower MDA contents than the wild type (WT) (Figure 11C–E). The results showed that the knockdown of *OsWRKY61* improved the cold tolerance of rice seedlings.

## 3. Discussion

The ceRNA hypothesis has been widely accepted since it was reported. In the ceRNA network, miRNAs are mainly responsible for the connection and regulation between different mRNAs and lncRNAs. mRNAs can be translated into proteins with direct functions, while lncRNAs that do not have translational functions indirectly affect mRNA expression through competitive binding with common miRNAs [41]. At present, there are many studies on mRNA in rice, but there is no report on the ceRNA network in response to low-temperature stress in rice. In this study, WGCNA was used to identify key modules and hub genes related to the low-temperature stress response, and it was found that SOD and POD had the highest correlation with the gene modules. Through OPLS-DA analysis, the VIP scores of the SOD and POD indexes were both greater than 1 (Figure 1), which were important physiological indexes of the antioxidant. The reliability of the WGCNA results is further proved. After that, we analyzed the hub genes and pathway significantly enriched genes in the module and constructed the ceRNA network.

The hub genes in the POD significantly related module included an *OsWRKY61* TF. Many WRKY proteins respond to various abiotic stresses [42,43]. When both rice varieties with opposite cold tolerance were sequenced for the transcriptome and proteome of low-temperature stress at gestation, this gene was differentially expressed, and it was significantly highly expressed in cold-sensitive rice varieties. Thus, this gene is a TF that may negatively regulate the cold tolerance of rice [44]. In this study, the expression was up-regulated in the cold-sensitive variety and down-regulated in the cold-tolerant variety, JL, further demonstrating the negative regulation of the *OsWRKY61* on cold tolerance in rice. In the ceRNA network, *OsWRKY61* is the target gene of *osa-miR5082*, interacting with *LNC3272*, *LNC1242*, and *LNC1229* (Figure 8). One plasma membrane hydrogen-ATPase (*Os02g0825600*) related to the ABA signal transduction pathway was included in the hub genes. The role of ABA in low-temperature stress has been confirmed in many plants [45,46]. Cold stress triggers ABA biosynthesis, as displayed by ABA-deficient and insensitive mutants [47,48]. Recent major advances in elucidating ABA signaling have revealed that the WRKY TF is a key node in the ABA response signaling network [49]. In the study of low-temperature stress in cucumber, *CsWRKY46* may regulate plant resistance to low temperature through the ABA-dependent signaling pathway [50]. In the study of low-temperature stress in rape, *BcWRKY46* was found improved plant low temperature tolerance by activating related genes in ABA signaling pathway [51]. In the study of low-temperature stress in grape, it was found that *VvWRKYs* and ABA-related genes were clustered in a co-expression network module, and it was speculated that the WRKY transcription factors might resist low-temperature injury by responding to exogenous sugars through the ABA signaling network [52]. In this study, O*sWRKY61* and ABA-related genes were also found in a network module. We inferred that the hub gene *OsWRKY61* in this study may also rely on ABA signaling networks to regulate the cold tolerance of rice seedlings, but this needs to be verified via experiments, which provides a new idea for our future research. We will confirm this finding through subsequent experiments.

The hub genes in the POD significantly related module included the UDP-glucosyltransferase family protein (*Os05g0527000*). With the extension of the low-temperature stress time, the expression of this gene in the two varieties continued to increase. It was found that overexpression of UDP-glucosyltransferase gene could enhance the low-temperature tolerance of rice [53]. One sesquiterpene synthase gene (*Os08g0167800*) was included in the hub genes. Sesquiterpenes play crucial roles in enhancing biotic or abiotic resistance to stress [54]. One IAA-related indole-3-glycerophosphoric synthetase gene (*Os03g0797500*) was included in the hub genes, and the expression of this gene increased significantly in the two varieties during 0–24h of low-temperature stress. IAA plays a crucial role in plant stress resistance, and its content is related to plants’ low-temperature tolerance [55]. In this study, the gene may regulate the synthesis of IAA to enable plants respond to low-temperature stress.

After that, we also constructed ceRNA networks for the significantly enriched pathway genes in the SOD- and POD-related modules. The 12-oxophytodienoate reductase (*Os06g0215900*), a precursor of jasmonic acid (JA) synthesis in the linoleic acid metabolic pathway, was found to be the target gene of *Novel.65*, interacting with *LNC0595* (Figure 8). JA is a ubiquitous plant signaling compound [56]. Exogenous JA can sustainably enhance the cold tolerance of *Arabidopsis thaliana* in response to low-temperature stress. Endogenous JA biosynthesis can also be activated by low-temperature stress, and JA positively regulates the C-repeat binding factor (CBF) transcription pathway, up-regulates downstream cold response genes, and ultimately, enhances cold tolerance [57]. Fatty acid alpha-dioxygenase (*Os12g0448900*) in the linoleic acid metabolic pathway is the target gene of osa *miR-156f-3p* and *Novel.65*, interacting with *LNC059*5, *LNC1508* and *LNC2976* (Figure 8). *miR156f-3p* has been shown to respond to salt stress in cotton [58]. Receptor-like kinase (*Os02g0615800*) in the linoleic acid metabolic pathway is the target gene of *osa-miR390-5p* and interacts with *LNC6350* (Figure 8). Receptor-like kinases play crucial roles in various abiotic stresses [59], including low-temperature stress in plants [60]. *osa-miR390-5p* has also been shown to be down-regulated in rice panices after low-temperature stress [61]. This suggests that *osa-miR390-5p* may be the key miRNA in rice’s response to low-temperature stress.

Finally, we performed subcellular localization analysis of *OsWRKY61* and found that it was located in the nucleus and chloroplast, and further gene knockout was performed to construct the mutant. It was found that *OsWRKY61* could reverse regulate the cold tolerance of rice plants, which was consistent with the results of previous functional studies on *OsWRKY63* [62]. There was no significant difference between the *OsWRKY63* overexpression plant and the wild type. However, under low-temperature stress, the survival rate of the overexpression plant was significantly lower than that of the wild type. Intriguingly, the use of CRISPR/Cas9 technology to knock down *OsWRKY6*3 expression led to enhanced cold tolerance [62]. This study preliminarily demonstrated the function of *OsWRKY61*. The linkage relationship between *OsWRKY61* and the ABA signal transduction pathway and the identification of RNA function in ceRNA network where *OsWRKY61* is located will become the focus of future research. In addition, the first establishment of the ceRNA network in rice response to low temperature also laid a foundation for the future study of rice.

## 4. Materials and Methods

### 4.1. Plant Materials and Cold Treatment

This experiment was conducted in 2021 in an artificial intelligence climate chamber (FYS-20, Nanjing, China) at Northeast Agricultural University (Harbin, Heilongjiang, China, longitude: 126°22′–126°50′; latitude: 45°34′–45°46′ N). Jilin Sunset (JL, a cold-tolerant japonica rice variety) and Jinhe (JH, a low cold-sensitive japonica rice variety) were the experimental materials. The experimental design was to select healthy and whole seeds, and after disinfection and germination, sow them in a hydroponic nutrition box, and then move them to the artificial intelligence climate chamber for further cultivation. The culture conditions were 20 °C day/18 °C night, 12 h light/12 h night cycle, and 50% relative humidity. When the rice seedlings grew into three leaves and one heart, they were moved to a 4 °C artificial intelligence climate chamber for low-temperature treatment. Leaves from the JL and JH rice varieties were harvested at 0, 4, 12, 24, and 48 h following the initiation of cold treatment. Each time point had three replicates for each variety. Each replicate was composed of a pooled sample of leaves from three individual seedlings (Figure 1A), resulting in a total of 30 samples collected across both varieties (2 varieties × 5 time points × 3 replicates). These samples were immediately frozen in liquid nitrogen and stored at −80 °C until further analysis.

### 4.2. Determination of Phenotypic Indicators

The height of the rice was measured from the base to the top using a ruler. A Vernier caliper was used to measure the natural width of the curled leaves (Ln) and the width of the leaves after stretching and flattening (Lg), and then calculated according to the formula DLC = (Lg − Ln)/Lg × 100%.

### 4.3. Determination of Physiological Indicators

Nine physiological indicators were measured in this study. The leaf SOD activity was determined by its ability to inhibit the photochemical reduction of nitroblue tetrazolium at 560 nm [63]. Furthermore, the CAT activity was determined by measuring the H_2_O_2_ consumed at 240 nm [64]. The POD activity was estimated using the guaiacol colorimetric method, monitoring the absorption changes at 470 nm during guaiacol oxidation [65]. Additionally, the MDA content was determined using a modified thiobarbituric acid chromatographic method [66]. Finally, the proline, SS, SP, and OH^−^contents and T-AOC were measured using a kit (Suzhou Grace Biotechnology Co., Ltd., Nanjing, China).

### 4.4. Library Construction and Sequencing of mRNA

The total RNA was extracted from rice leaves using an RNA extraction kit (TIANGEN Biotech, Beijing, China); the concentration and integrity of the extracted RNA were measured using Nanodrop 2000 (Thermo Fisher Scientific, Waltham, MA, USA). The extracted total RNA was then subjected to rRNA removal using an rRNA removal kit (Epicentre Technologies, Madison, WI, USA). The first-strand cDNA was synthesized using PCR, followed by second-strand cDNA, end-repair and 3’ end addition A. Finally, sequencing libraries were generated according to the manufacturer’s standards and sequenced on the Illumina NovaSeq 6000 (Illumina, San Diego, CA, USA) platform.

### 4.5. Mapping and De Novo Assembly

To obtain clean data, we removed any low-quality and contaminated parts of the raw data. Subsequently, we performed sequence alignment of the clean data with the rice reference genome (http://rice.plantbiology.msu.edu/, accessed on 2 January 2024) via MSU-v7.0 on 1 April 2022, and we then assembled the reads on the alignment using StringTie [67]. Finally, annotation was performed using Cuffcompare software v2.2.1 [68].

### 4.6. Identifying Differentially Expressed lncRNAs and mRNAs

For the reconstituted transcripts, the lncRNA screening included (1) filtering transcripts shorter than 200 nt in length; (2) comparing sequences to the Rfam [69] database using BLAST (http://blast.ncbi.nlm.nih.gov, accessed on 2 January 2024) to filter small non-coding RNAs on 7 April 2022, such as tRNA, snRNA; (3) performing a structural domain search of the Pfam [70] database using hmmsearch [71] to filter sequences with known protein structural domains; and (4) coding the ability prediction of the remaining transcripts using CPC2 [72]. Finally, the lncRNAs were considered reliable by screening. Read counts of all the lncRNAs and mRNAs were performed using StringTie software, with FPKM as a measure of lncRNA and mRNA expression levels. DEmRNAs and DElncRNAs were identified using DESeq2 (http://www.bioconductor.org/, accessed on 2 January 2024) based on the criteria of a fold change ≥ 2 and FDR ≤ 0.01 on 10 April 2022.

### 4.7. Target Gene Prediction of DElncRNAs

The function of lncRNA is mainly realized by acting on protein-coding target genes in cis or trans mode. Therefore, the main functions of lncRNA can be predicted by the function of lncRNA target genes. Based on the basic principle of cis target gene prediction, it is believed that the function of lncRNA is related to the protein-coding genes adjacent to its coordinate, so the protein-coding genes in the vicinity of lncRNA (100 k range in the upstream and downstream) were screened as its cis target genes. Based on the basic principle of trans target gene prediction, it is believed that the function of lncRNA does not depend on the position relationship with the coding gene but is related to the protein coding gene co-expressed with lncRNA. The target genes can be predicted via the correlation analysis of lncRNA and protein-coding gene expression between samples. The Pearson correlation coefficient method was used to analyze the correlation between the lncRNA and protein-coding genes among the samples. Protein-coding genes with high correlation (r > 0.9, *p* < 0.01) were selected as trans target genes. After that, KEGG enrichment analysis was performed for all the lncRNA target genes.

### 4.8. Library Construction and Sequencing of miRNA

The libraries were constructed using the TruSeq Small RNA Sample Prep Kit (Illumina, San Diego, CA, USA), the concentrations of the libraries were checked using Qubit 2.0 (Thermo Fisher Scientific), and the insert size was checked using an Agilent 2100 bioanalyzer (Agilent Technologies, Santa Clara, CA, USA). The effective concentrations of the libraries were accurately quantified using the Q-PCR method. After the library test, high-throughput sequencing was performed using HighSeq (Illumina).

To ensure the accuracy of the information analysis, quality control of the raw data was required, with the following criteria: (1) for each sample, sequences with low-quality values were removed; (2) reads with an unknown base N content greater than or equal to 10% were removed; (3) reads without 3’ junction sequences were removed; (4) 3’ junction sequence; and (5) sequences shorter than 18 or longer than 30 nucleotides were removed. The clean reads were sequenced against the Silva database (http://www.arb-silva.de, accessed on 2 January 2024), the GtRNAdb database (http://gtrnadb.ucsc.edu/, accessed on 2 January 2024), the Rfam database, and the Repbase database (http://bioinfo-tool.cp.utfpr.edu.br/plantemirdb/, accessed on 2 January 2024) to filter out rRNA, tRNA, snRNA, and snoRNA on 7 May 2022, respectively. The unannotated reads were compared with the rice reference genome (http://plants.ensembl.org/Oryza_sativa/Info/Index, accessed on 2 January 2024) using the Bowtie software [73] for sequence alignment and the miRDeep-P2 [74] for predicting new miRNAs on 10 May 2022.

### 4.9. Analysis of miRNA Expression and Differential Identification

The miRNAs in each sample were counted and normalized for expression using the TPM algorithm, followed by differential expression analysis using DESeq2 based on a fold change ≥ 1.5 and *p*-value ≤ 0.05 as the screening criteria.

### 4.10. Weighted Gene Co-Expression Network Analysis

First, the gene FPKM values were imported into the WGCNA R package complex [75] and the topological overlap matrix (TOM) was used to construct the WGCNA network. Then, the genes were clustered using hclust, the clustering results were cut using the dynamic tree-cutting method, and the correlation coefficients of the divided modules with the physiological indicators were calculated to find the significantly correlated modules.

### 4.11. GO and KEGG Enrichment Analysis

The enrichment analysis of RNA within significantly related modules in the WGCNA analysis was performed using topGO (http://www.bioconductor.org/packages/release/bioc/html/topGO.html) on 20 May 2022. GO terms with a *p*-value ranking of 10 were considered significantly enriched for mRNAs. Statistical enrichment of RNA within the modules was performed using the R package detection Kyoto Encyclopedia of Genes and Genomes (KEGG). The KEGG pathways were ranked according to the corrected *p*-values, and the top 10 pathways were considered the most enriched.

### 4.12. ceRNA Network Construction and Analysis

To explore the synergistic functions of DElncRNAs, we constructed the ceRNA network. First, the target genes of differential miRNAs and lncRNAs were predicted using TargetFinder (www.ambion.com/techlib/misc/siRNA_Finder.html) on 27 May 2022, which is a pair of miRNA–mRNA and miRNA–lncRNA target gene relationships. Subsequently, mRNAs in the module significantly associated with phenotypic traits were intersected with mRNAs in the differential miRNA–mRNA relationship pairs to obtain the intersected differentially expressed miRNA–mRNA relationship pairs. Then, for an lncRNA–mRNA pair, mRNAs and lncRNAs that serve as common miRNA targets and are negatively co-expressed with their miRNAs were selected as co-expression competitive triplets. Finally, the data were visualized using Cytoscape 3.3.0 software (http://www.cytoscape.org) on 28 May 2022.

### 4.13. RNA Extraction and qRT-PCR

Rice leaves were treated at 4 °C for 0 h, 4 h, 12 h, 24 h, and 48 h, and the total RNA was extracted using a TranZol Up RNA kit (TianGen Biotech, Beijing, China). Reverse transcription was performed using a HiFiScript cDNA Synthesis Kit (Cwbio, Beijing, China), and qRT-PCR was performed on the Roche Light Cycler 2.10 system with three technical replicates and three biological replicates per sample. Actin1 was used as a reference gene, and the relative expression level of the gene was calculated using 2^−ΔΔCt^.

### 4.14. Subcellular Localization of OsWRKY61

The full-length CDS sequence of *OsWRKY61* without the stop codon was amplified and then inserted into a modified pCambia1300-GFP vector with a CaMV35S promoter. The obtained GFP-*OsWRKY61* fusion vector was transformed into *Arabidopsis* protoplasts and incubated in the dark at 28 °C for 16 h. Finally, the GFP fluorescence was measured via fluorescence microscopy (Zeiss LSM710; Carl Zeiss, Oberkochen, Germany).

### 4.15. Construction, Screening, and Determination of Cold Tolerance of OsWRKY61 Mutant Plants

We selected two 23 bp target sites (GGTCGACGTCGTATACAAGG GGG, CCAGCTGTAACCGTCGTCTG GGG) on the exon of *OsWRKY61* and designed a pair of target junction primers: ZH-F (ACCCAAAGTAAGTAAGCGATGA) and ZH-R (CGAGATCAGTTCGATTTCATGT). The specificity of the target gRNAs was evaluated using the biology website CRISPR-GE (http://skl.scau.edu.cn/, accessed on 2 January 2024) and the NCBI (https://blast.ncbi.nlm.nih.gov) on 7 August 2022. The target sequence had at least two bases different from similar off-target sequences within the PAM or PAM-proximal region. Then, the target site was integrated into the gene-editing vector *PEGCas9PUB-H*, and the following steps were followed. (1) 1 µL plasmid was added to 50 µL receptive cells of *Agrobacterium tumefaciens* EHA105, thoroughly mixed and absorbed into a spinning cup, 1 mL LB liquid medium was added after spinning, thoroughly mixed and absorbed into a 1.5 mL centrifuge tube, and incubated in a shaking table at 30 °C and 180 rpm for 30 min. After that, 50 µL of the activated *Agrobacterium tumefaciens* solution was absorbed and inoculated on LB solid medium and incubated at 30 °C for 48 h. (2) The rice grains with no mildew and normal bud were selected, disinfected with 75% alcohol for 3 times, and washed with distilled water for 1 min/time. Then, the rice grains were disinfected with 15% sodium hypochlorite for 20 min, washed with distilled water 3 times, and finally inoculated into induction medium and cultured under light at 26 °C for 20 days. (3) *Agrobacterium tumefaciens* was selected in the infection solution to prepare the *Agrobacterium tumefaciens* suspension with OD600 = 0.2, and then the callus was selected in a triangular bottle, the *Agrobacterium tumefaciens* suspension was added, and the suspension was discarded after infection for 10–15 min. The callus was inoculated in the co-culture medium and co-cultured at 20 °C for 48–72 h. (4) The callus after infection was inoculated on screening medium and incubated at 26 °C for 20–30 days. (5) The positive callus was inoculated into the secondary screening medium, and monoclonal callus must be selected during callus selection, and dark culture at 26 °C for 7–10 days. (6) The positive callus was inoculated into the differentiation medium, cultured using light at 25–27 °C for 15–20 days, and then inoculated into the rooting medium after the buds of 2–5 cm were differentiated. Finally, the plants were transplanted to the field for planting and management, and the homozygous T3 generation regeneration plants were screened via PCR amplification and sequencing. The screened plants of different mutant types were cultured together with the wild-type plants until three leaves and one heart according to the above method and then subjected to low-temperature treatment at 4 °C. Samples were taken after 48 h of cold treatment to determine the SOD, proline and MDA contents according to the above method.

## 5. Conclusions

In this experiment, the physiological indexes of two japonica rice varieties with significantly different cold tolerance were measured under low temperatures. The antioxidant enzyme activities and osmotic substances of the cold-tolerant varieties were regulated for a prolonged time and were relatively stable, which could remove harmful substances from the body and promptly maintain plant cells’ stability. Based on the VIP, SOD, POD and OH^−^were identified as the most important antioxidant phenotypes. Combining the antioxidant physiological indicators with the entire transcriptome data via WGCNA further found that the gene modules had the highest correlation with key antioxidant enzymes SOD and POD. Enrichment analysis revealed that the SOD-related module genes were significantly enriched in the redox pathway, with the hub genes of this module including *Os05g0527000*, *Os08g0167800* and *Os03g0797500*. The POD-related module genes were significantly enriched in the linoleic acid metabolic pathway, and the module hub genes included *Os11g0685700*, *Os02g0825600*, and *Os02g0615800*. The module hub genes and module significantly enriched pathway genes were selected to construct a key competitive endogenous ceRNA network. The network consisted of three mRNAs, four miRNAs, and seven lncRNAs. *LNC1608*, *LNC0595*, *LNC6350*, *LNC1242*, *osa-miR156f-3p*, *Novel.65* and *osa-miR390-5p* were found to be the key candidate genes for cold tolerance. Finally, subcellular localization and mutant validation of the candidate gene *OsWRKY61* confirmed the involvement of this gene in regulating cold tolerance in rice.

## Figures and Tables

**Figure 1 ijms-25-02310-f001:**
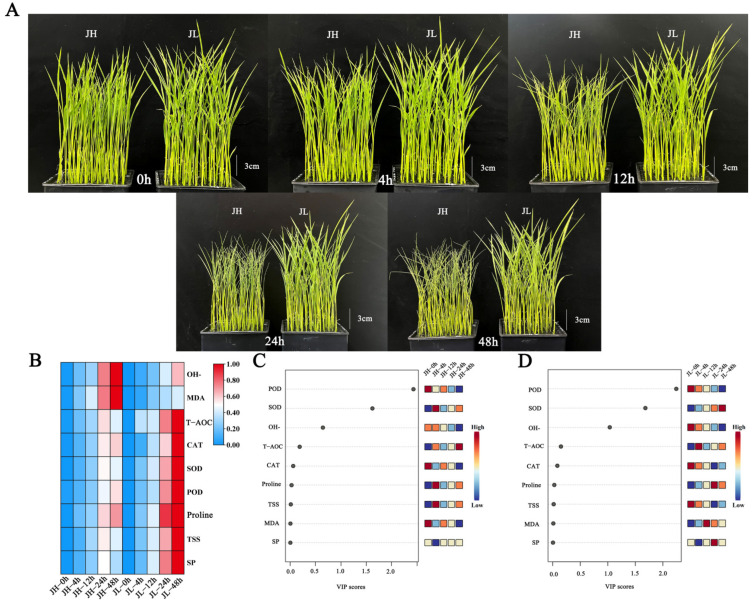
Morphological and physiological characteristics of rice leaves after five time points of cold treatment. (**A**) Phenotypic characteristics of rice seedling leaves after 0 h, 4 h, 12 h, 24 h and 48 h of cold treatment. (**B**) Heat map of 9 physiological index data after 0 h, 4 h, 12 h, 24 h and 48 h of cold treatment. (**C**) VIP scores of nine physiological indices of JH after five time points of cold treatment. (**D**) VIP scores of nine physiological indices of JL after five time points of cold treatment.

**Figure 2 ijms-25-02310-f002:**
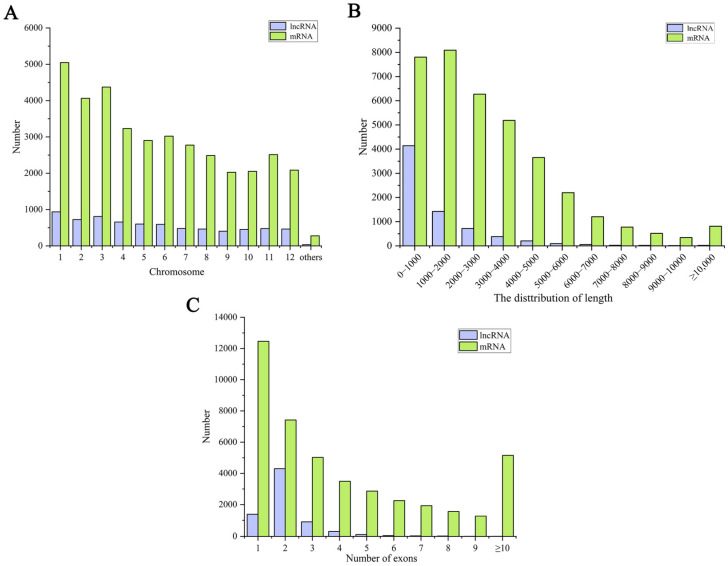
Information on the identified mRNAs and lncRNAs. (**A**) The number of the identified mRNAs and lncRNAs on chromosomes. (**B**) The length number of the identified mRNAs and lncRNAs. (**C**) The exon number of the identified mRNAs and lncRNAs.

**Figure 3 ijms-25-02310-f003:**
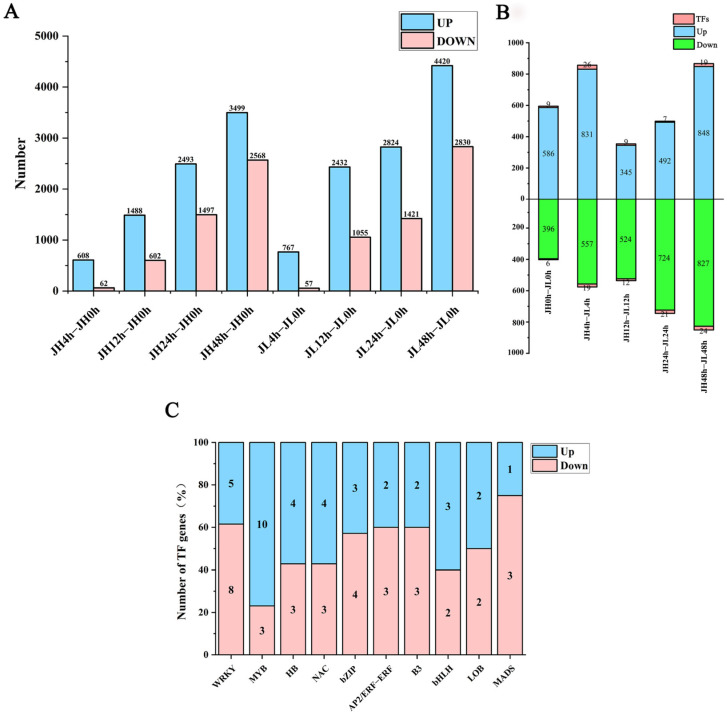
Differential analysis of mRNAs in leaves of rice seedlings. (**A**) The number of differential genes of JH and JL at each time point after cold treatment. (**B**) The number of differentially up- and down-regulated genes in JL at each time point after cold treatment compared to JH. (**C**) Number of significantly highly expressed and significantly lowly expressed TFs in the differential transcription factors.

**Figure 4 ijms-25-02310-f004:**
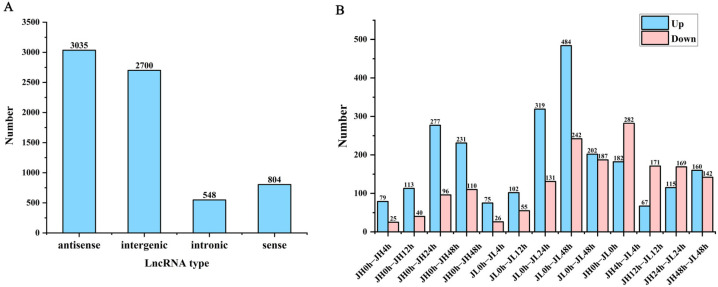
Classification and differential analysis of lncRNAs in rice seedling leaves. (**A**) Number of rice lncRNAs classified according to genomic location. (**B**) Number of differential lncRNAs between JH and JL at each time point after cold treatment and the number of differential lncRNAs in JL compared to JH at each time point after cold treatment.

**Figure 5 ijms-25-02310-f005:**
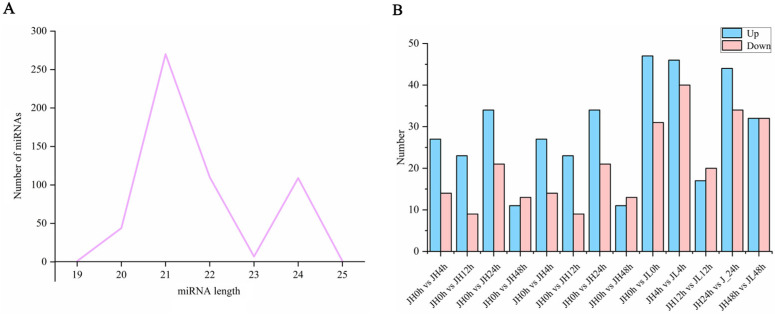
Identification and differential analysis of miRNAs. (**A**) Length distribution of the identified miRNAs. (**B**) Number of differential miRNAs between JH and JL at each time point after cold treatment and the number of differential miRNAs in JL compared to JH at each time point after cold treatment.

**Figure 6 ijms-25-02310-f006:**
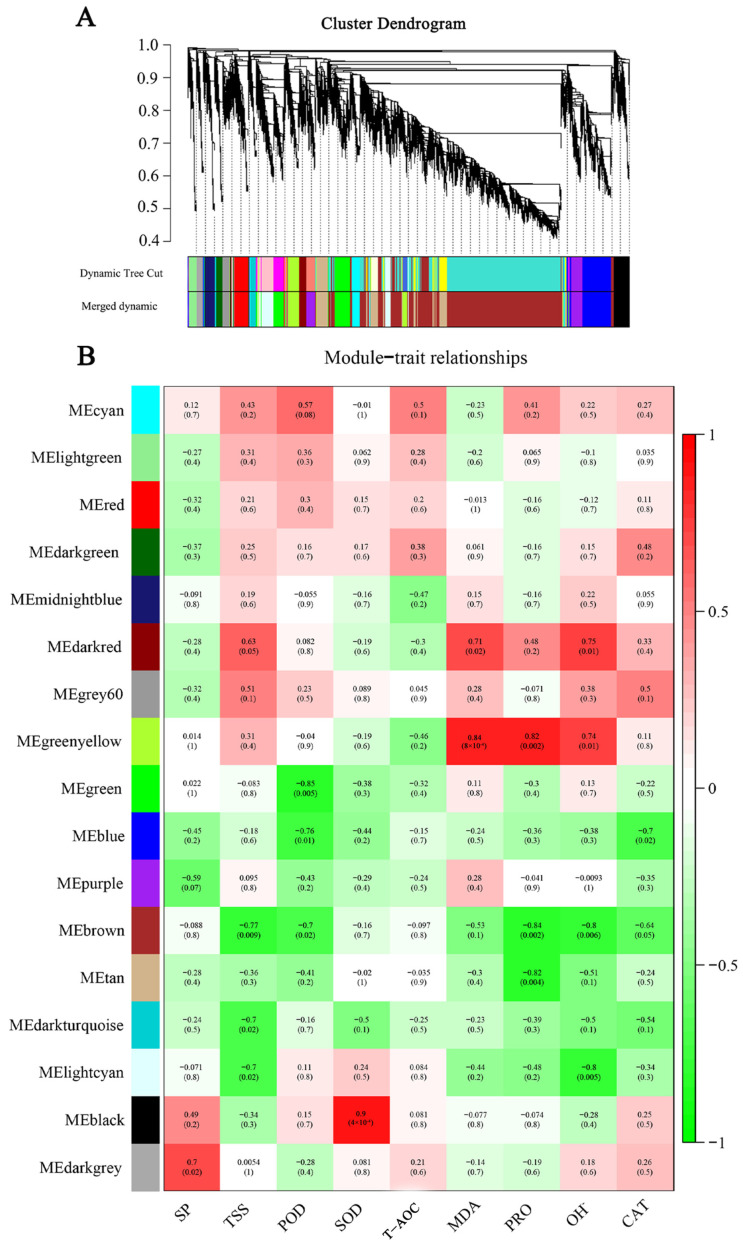
WGCNA of efficiently expressed genes in two varieties under low-temperature stress. (**A**) Hierarchical clustering tree of co-expression modules identified by WGCNA. (**B**) Correlation and corresponding *p*-values between modules and phenotypic traits after low temperature treatment.

**Figure 7 ijms-25-02310-f007:**
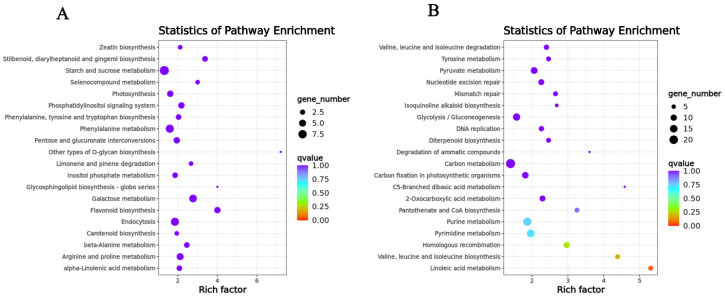
KEGG enrichment analysis of genes in the key co-expression modules. (**A**) KEGG enrichment analysis of genes contained in the SOD-related module. (**B**) KEGG enrichment analysis of genes contained in the POD-related module.

**Figure 8 ijms-25-02310-f008:**
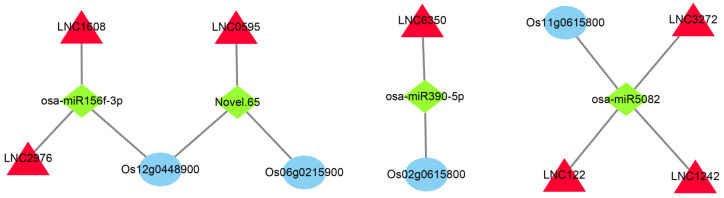
CeRNA network construction of the key co-expression modules. The red triangle represents lncRNA, the green quadrilateral represents miRNA, and the blue oval represents mRNA. Blue represents mRNAs, red denotes lncRNAs, and green indicates miRNAs.

**Figure 9 ijms-25-02310-f009:**
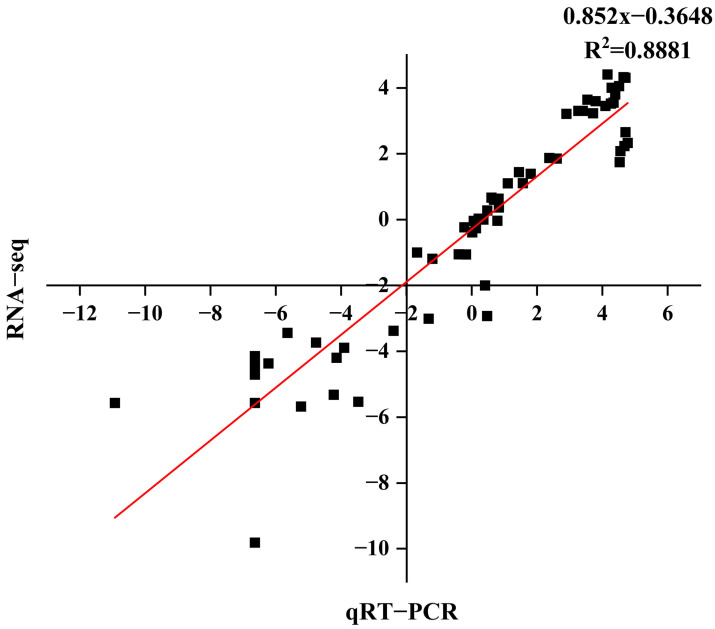
Correlation analysis of the DEGs between RNA-Seq and qRT-PCR.

**Figure 10 ijms-25-02310-f010:**
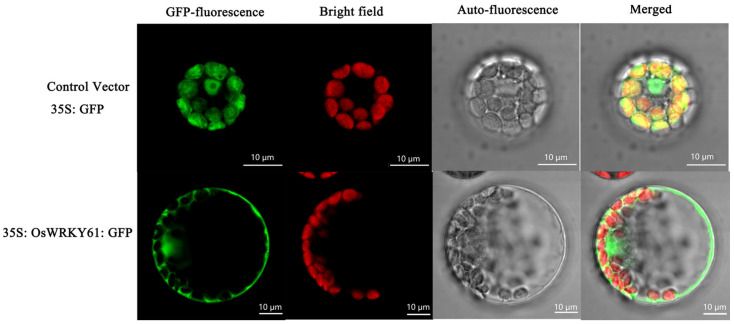
Subcellular localization analysis of *OsWRKY61*.

**Figure 11 ijms-25-02310-f011:**
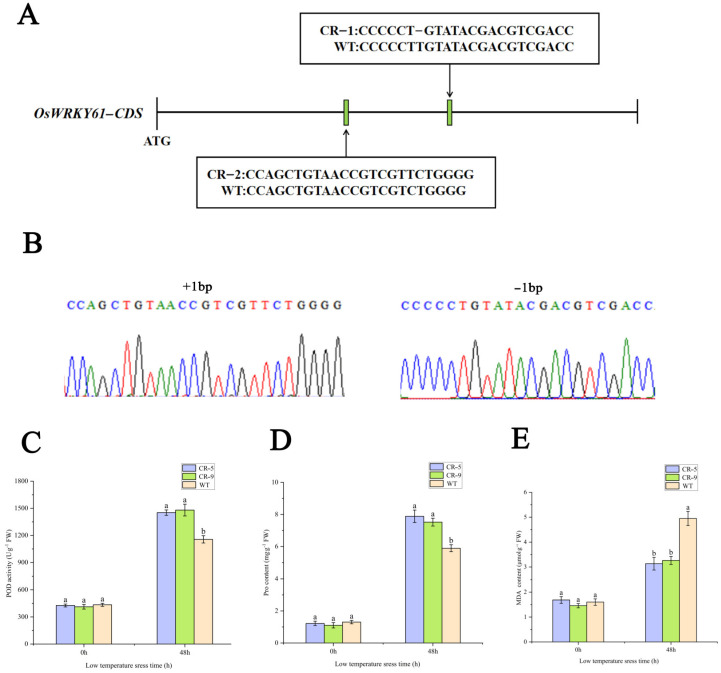
Construction of *OsWRKY61* mutant and determination of physiological indexes after cold treatment. (**A**) Schematic diagram of *OsWRKY61* mutation types. (**B**) Sequencing map of *OsWRKY61* mutant. (**C**–**E**) SOD, proline and MDA content changes in the *OsWRKY61* mutants and WT after cold treatment. Labels “a” and “b” indicate statistical.

## Data Availability

The original data presented in this study are included in both the article and its Appendix A. The raw sequences analyzed in this study can be found in the corresponding online repositories under the accession numbers BioProject ID: PRJNA1071886 and BioProject ID: PRJNA1068025. For further inquiries, please direct your questions to the corresponding author(s).

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
