# Peer review of "Whole-Transcriptome Profiling and Functional Prediction of Long Non-Coding RNAs Associated with Cold Tolerance in Japonica Rice Varieties"

_ijms, 2024, doi:10.3390/ijms25042310_

Round 1

Reviewer 1 Report

Comments and Suggestions for Authors

In this study, the authors integrated antioxidant physiological indices with whole transcriptome data using weighted gene co-expression network analysis (WGCNA) to identify genes associated with cold tolerance in rice. Furthermore, the negative regulatory function of OsWRKY61 was determined. The findings of this research hold significant scientific value and it is recommended for publication.

1, Some formats should be consistent above and below, such as 2.12. Mutant construction and phenotype analysis should be in italics.

2, The differences shown in Figure 1A are not described and are suggested to be supplemented. 

Author Response

For research article

Response to Reviewer 1 Comments

1. Summary

Thanks for your comments concerning our manuscript entitled " Whole-Transcriptome Profiling and Functional Prediction of Long Non-Coding RNAs Associated with Cold Tolerance in Japonica Rice Varieties". These comments are valuable and helpful for us to revise our manuscript and guide our future research. We had carefully reviesed the paper according to these comments, and we hope this revision would be approved by you. The main changes in the manuscript and responses to these comments are listed below.

2. Questions for General Evaluation

Reviewer’s Evaluation

Response and Revisions

Does the introduction provide sufficient background and include all relevant references?

Yes/Can be improved/Must be improved/Not applicable

Are all the cited references relevant to the research?

Yes/Can be improved/Must be improved/Not applicable

Is the research design appropriate?

Yes/Can be improved/Must be improved/Not applicable

Are the methods adequately described?

Yes/Can be improved/Must be improved/Not applicable

Are the results clearly presented?

Yes/Can be improved/Must be improved/Not applicable

Are the conclusions supported by the results?

Yes/Can be improved/Must be improved/Not applicable

3. Point-by-point response to Comments and Suggestions for Authors

Comments 1: Some formats should be consistent above and below, such as “2.12. Mutant construction and phenotype analysis“ should be in italics.

Response 1: Thank you very much for your sincere suggestion. We have corrected the format of this sentence as you suggested, making it italicized. (line 404)

Comments 2: The differences shown in Figure 1A are not described and are suggested to be supplemented.

Response 2: Agree. I/We have, accordingly, revised all images appearing in the manuscript, such as lines 128, 149, 189, 216, and 241. We have improved the resolution of these images, and some of them have been remade and improved.   Thank you very much for your sincere suggestion. We have added description of Figure 1A as you suggested. (lines 132-140)

Reviewer 2 Report

Comments and Suggestions for Authors

I recommend don’t include gene numbers in abstract (such as Os05g0527000) as well as abbreviations of names that are not further mentioned in abstract (TF, mRNAs, miRNAs, lncRNAs). Include this abbreviation in main text (line 45 – TF). Information about black and green modules is not clear in abstract. As I understand it, any of the modules may be interesting for research, and the colors of the modules themselves are not universal in different studies.

Add information about OsCNGC (that it is a cyclic nucleotide-gated channel).

Check that genes and species are written in italics (eg gene CML27, MuLnc1, Dickeya zeae, Arabidopsis thaliana - lines 91-108; Oryza sativa line 190).

Line 130 Please add information about varieties of rice and add a link to Figure 1A there.

Please correct the size of the pictures based on the size of the text in the pictures and in the body text.

What programs did you use to remove low-quality and contaminated parts of the original data?

Please, add a scale bar to Figure 1A.

I’m not sure that different amounts of mRNA and lncRNA may lead to the conclusion that “These results suggest that mRNAs and lncRNAs play different roles in biological pathways.”

Add links to TargetFinder, topGO, Cytoscape 3.3.0.

Please, add abbreviation decoding of DEmRNAs, DEGs, DElncRNA.

Add coordinates and nucleotide sequences of 113 novel miRNAs. Why in the Table S3 133 Novel miRNA? If you have identified more new RNAs, please also indicate their coordinates and sequences.

Which DEGs were selected for polymerase chain reaction (qRT-PCR) to validate the RNA-Seq results? Please provide list of primers.

Line 372 Please add a description of why OsWRKY61 was selected for further analysis.

There is no caption for Figure 10.

Line 585 Please add a plant transformation protocol.

Is the growth of JH rice significantly reduced when exposed to cold? Does such a decrease occur in mutant lines? Could you please add morphological characteristics of mutant lines?

In conclusion – “By selecting module hub genes and module significantly enriched pathway genes to construct key competitive endogenous ceRNA networks” – add verb.

Many parts of the text are missing introductory sentences, and some of the sentences are out of place (For example: lines 49-51 «A novel class of gene regulatory molecules, lncRNAs have gained more attention to further explore the regulatory mechanism of plants under cold stress [10].»). The most confusing part is the "discussion" section, which I highly recommend rewriting.

Author Response

For research article

Response to Reviewer 2 Comments

1. Summary

Thanks for your comments concerning our manuscript entitled " Whole-Transcriptome Profiling and Functional Prediction of Long Non-Coding RNAs Associated with Cold Tolerance in Japonica Rice Varieties". These comments are valuable and helpful for us to revise our manuscript and guide our future research. We had carefully reviesed the paper according to these comments, and we hope this revision would be approved by you. The main changes in the manuscript and responses to these comments are listed below.

2. Questions for General Evaluation

Reviewer’s Evaluation

Response and Revisions

Does the introduction provide sufficient background and include all relevant references?

Yes/Can be improved/Must be improved/Not applicable

Are all the cited references relevant to the research?

Yes/Can be improved/Must be improved/Not applicable

Is the research design appropriate?

Yes/Can be improved/Must be improved/Not applicable

Are the methods adequately described?

Yes/Can be improved/Must be improved/Not applicable

Are the results clearly presented?

Yes/Can be improved/Must be improved/Not applicable

Are the conclusions supported by the results?

Yes/Can be improved/Must be improved/Not applicable

3. Point-by-point response to Comments and Suggestions for Authors

Comments 1: I recommend don’t include gene numbers in abstract (such as Os05g0527000) as well as abbreviations of names that are not further mentioned in abstract (TF, mRNAs, miRNAs, lncRNAs). Include this abbreviation in main text (line 45 – TF). Information about black and green modules is not clear in abstract. As I understand it, any of the modules may be interesting for research, and the colors of the modules themselves are not universal in different studies.

Response 1: Thank you very much for your sincere suggestion. We have deleted the gene number and abbreviations that are not further mentioned in the abstract according to your suggestion. We have also modified the problem you raised about the unclear information of the black module and the green module to make it clearer. (lines 16-20, 26, 44)

Comments 2: Add information about OsCNGC (that it is a cyclic nucleotide-gated channel).

Response 2: Thank you very much for your sincere suggestion. We have added the information about OsCNGC. (line 46)

Comments 3: Check that genes and species are written in italics (eg gene CML27, MuLnc1, Dickeya zeae, Arabidopsis thaliana - lines 91-108; Oryza sativa line 190).

Response 3: Thank you very much for your sincere suggestion. We have checked and corrected the full text of genes and species are written in italics. (lines 44, 93, 101, 103, 104, 107, 110)

Comments 4: Line 130 Please add information about varieties of rice and add a link to Figure 1A there.

Response 4: Thank you very much for your sincere suggestion. We have add information about varieties of rice and add a link to Figure 1A there. (lines 121-122)

Comments 5: Please correct the size of the pictures based on the size of the text in the pictures and in the body text.

Response 5: Thank you very much for your sincere suggestion. We have corrected the size of the pictures based on the size of the text in the pictures and in the body text. (lines 189, 221, 251, 288, 303, 320, 361, 378, 388, 402, 421)

Comments 6: What programs did you use to remove low-quality and contaminated parts of the original data?

Response 6: Thank you very much for your sincere suggestion. We use Trimmomatic software to remove low quality and contaminated data and incorporated the software into the manuscript as you suggested. (lines 121-122)

Comments 7: Please, add a scale bar to Figure 1A.

Response 7: Thank you very much for your sincere suggestion. We have added a scale bar to Figure 1A. (line 189)

Comments 8: I’m not sure that different amounts of mRNA and lncRNA may lead to the conclusion that “These results suggest that mRNAs and lncRNAs play different roles in biological pathways.”

Response 8: Thank you very much for your sincere suggestion. We have changed this conclusion in the text based on your suggestion. (lines 219-220)

Comments 9: Add links to TargetFinder, topGO, Cytoscape 3.3.0.

Response 9: Thank you very much for your sincere suggestion. We have added links to TargetFinder, topGO, Cytoscape 3.3.0. (lines 608, 599, 616)

Comments 10: Please, add abbreviation decoding of DEmRNAs, DEGs, DElncRNA.

Response 10: Thank you very much for your sincere suggestion. We have added abbreviation decoding of DEmRNAs, DEGs, DElncRNA. (lines 370-371, 382, 281-282)

Comments 11: Add coordinates and nucleotide sequences of 113 novel miRNAs. Why in the Table S3 133 Novel miRNA? If you have identified more new RNAs, please also indicate their coordinates and sequences.

Response 11: Thank you very much for your sincere suggestion. We have corrected the mistake of writing 133 as 113, and added the coordinates and nucleotide sequences of 133 novel miRNAs in Table S2 according to your requirements. (line 296)

Comments 12: Which DEGs were selected for polymerase chain reaction (qRT-PCR) to validate the RNA-Seq results? Please provide list of primers.

Response 12: Thank you very much for your sincere suggestion. According to your suggestions, we have put the list of gene names and primers selected for qRT-PCR in Table S6. (lines 385-386)

Comments 13: Line 372 Please add a description of why OsWRKY61 was selected for further analysis.

Response 13: Thank you very much for your sincere suggestion. We have added the description of why OsWRKY61 was selected for further analysis. (lines 391-395)

Comments 14: There is no caption for Figure 10.

Response 14: Thank you very much for your sincere suggestion. We have added a caption to Figure 10. (line 403)

Comments 15: Line 585 Please add a plant transformation protocol.

Response 15: Thank you very much for your sincere suggestion. We have add a plant transformation protocol according to your requirements. (lines 635-638)

Comments 16: Is the growth of JH rice significantly reduced when exposed to cold? Does such a decrease occur in mutant lines? Could you please add morphological characteristics of mutant lines?

Response 16: Thank you very much for your sincere suggestion. The growth of JH rice is significantly reduced under low temperature conditions, and we have added a description of the growth reduction in the article based on your suggestion.(lines 132-140). As for the photos of the mutant strain, due to the loss of the photos due to the damage of our storage equipment, we cannot add pictures temporarily, so we have tested the physiological indexes of the mutant cold tolerance to prove it, but your opinion is very meaningful, we will pay attention to this point in the future publication of the article, and we will also back up the relevant photos, and we sincerely thank you for your suggestion

Comments 17: In conclusion – “By selecting module hub genes and module significantly enriched pathway genes to construct key ceRNA networks” – add verb.

Response 17: Thank you very much for your sincere suggestion. We have added verb to this sentence based on your suggestions. (lines 657-658)

Comments 18: Many parts of the text are missing introductory sentences, and some of the sentences are out of place (For example: lines 49-51 «A novel class of gene regulatory molecules, lncRNAs have gained more attention to further explore the regulatory mechanism of plants under cold stress [10].»). The most confusing part is the "discussion" section, which I highly recommend rewriting.

Response 18: Thank you very much for your sincere suggestion. We have corrected the sentence in the introduction of the whole article according to your suggestion, and also rewritten the discussion part of the article. (lines 427-511)

Reviewer 3 Report

Comments and Suggestions for Authors

The manuscript entiled "Whole-Transcriptome Profiling and Functional Prediction of 2 Long Non-Coding RNAs Associated with Cold Tolerance in Ja- ponica Rice Varieties" by Wang et al., investigates the effects of low-temperature chilling on rice, a crop sensitive to cold stress and important for food security. The authors measured antioxidant physiological indices, analyzed whole transcriptome data using WGCNA, identified gene modules correlated with antioxidant physiological indices, and constructed competitive endogenous networks involving long non-coding RNAs, miRNAs, and mRNAs. They also validated the negative regulatory role of a WRKY transcription factor (OsWRKY61) in cold stress response.

The manuscript to me seems to be an interesting study, however, they are some of my concerns in reference to the current version of the manuscript:

    1. The meaning of "PRO" should be explicitly provided in the manuscript if it is an abbreviation. The full name or meaning of "PRO" is necessary for the readers to understand the context.

    2. jL and jH Abbreviations: Similarly, any abbreviations such as jL and jH should be defined when they are first used in the manuscript to ensure clarity and understanding for the readers.

    3. What experimental design does authors have used in their study, it should be provided in detail in the method section? The experimental design used in the study should be detailed in the methods section to provide a clear understanding of the methodology employed by the authors.

    4. Number of Samples and Replicates: The number of samples and replicates used for physiological studies and transcriptome sequencing should be explicitly mentioned in the manuscript. This information is crucial for assessing the robustness and reliability of the findings.

    5. Library Preparation: The manuscript should clarify whether whole libraries were prepared or if mRNA libraries were prepared. Since two library preparation protocols are mentioned in sections 4.3 and 4.6, it's important to specify which protocol was used for the study.

    6. The legend for Figure 10 is missing, and it's essential for providing context and understanding the content of the figure. The authors should ensure that all figures are accompanied by descriptive legends to convey the necessary information to the readers.

These concerns are valid for ensuring the clarity, transparency, and reproducibility of the study. If these details are not explicitly provided in the current version of the manuscript, it would be beneficial for the authors to address these points in a revised version to enhance the overall quality and comprehensibility of the research.

Comments on the Quality of English Language

The language to me seems to be fine, however author can check of some of the typos in the manuscript

Author Response

For research article

Response to Reviewer 3 Comments

1. Summary

Thanks for your comments concerning our manuscript entitled " Whole-Transcriptome Profiling and Functional Prediction of Long Non-Coding RNAs Associated with Cold Tolerance in Japonica Rice Varieties". These comments are valuable and helpful for us to revise our manuscript and guide our future research. We had carefully reviesed the paper according to these comments, and we hope this revision would be approved by you. The main changes in the manuscript and responses to these comments are listed below.

2. Questions for General Evaluation

Reviewer’s Evaluation

Response and Revisions

Does the introduction provide sufficient background and include all relevant references?

Yes/Can be improved/Must be improved/Not applicable

Are all the cited references relevant to the research?

Yes/Can be improved/Must be improved/Not applicable

Is the research design appropriate?

Yes/Can be improved/Must be improved/Not applicable

Are the methods adequately described?

Yes/Can be improved/Must be improved/Not applicable

Are the results clearly presented?

Yes/Can be improved/Must be improved/Not applicable

Are the conclusions supported by the results?

Yes/Can be improved/Must be improved/Not applicable

3. Point-by-point response to Comments and Suggestions for Authors

Comments 1: The meaning of "PRO" should be explicitly provided in the manuscript if it is an abbreviation. The full name or meaning of "PRO" is necessary for the readers to understand the context.

Response 1: Thank you very much for your sincere suggestion. We have added the PRO abbreviation to the article based on your suggestion. (line 65)

Comments 2: jL and jH Abbreviations: Similarly, any abbreviations such as jL and jH should be defined when they are first used in the manuscript to ensure clarity and understanding for the readers..

Response 2: Thank you very much for your sincere suggestion. We have added abbreviated definitions for JH and JL to the article based on your suggestion. (lines 121-122)

Comments 3: What experimental design does authors have used in their study, it should be provided in detail in the method section? The experimental design used in the study should be detailed in the methods section to provide a clear understanding of the methodology employed by the authors.

Response 3: Thank you very much for your sincere suggestion. We have added the experimental design in detail in the method section according to your suggestion. (lines 518-524)

Comments 4: Number of Samples and Replicates: The number of samples and replicates used for physiological studies and transcriptome sequencing should be explicitly mentioned in the manuscript. This information is crucial for assessing the robustness and reliability of the findings.

Response 4: Thank you very much for your sincere suggestion. We have added the number of samples and replicates used for physiological studies and transcriptome sequencing. (lines 524-529)

Comments 5: Library Preparation: The manuscript should clarify whether whole libraries were prepared or if mRNA libraries were prepared. Since two library preparation protocols are mentioned in sections 4.3 and 4.6, it's important to specify which protocol was used for the study.

Response 5: Thank you very much for your sincere suggestion. We have defined the type of library preparation as suggested by you. (lines 539, 566)

Comments 6: The legend for Figure 10 is missing, and it's essential for providing context and understanding the content of the figure. The authors should ensure that all figures are accompanied by descriptive legends to convey the necessary information to the readers.

Response 6: Thank you very much for your sincere suggestion. We have added a legend to Figure 10 as you suggested. (line 403)

Round 2

Reviewer 2 Report

Comments and Suggestions for Authors

Dear authors,

Thank you very much for your excellent work with the text.

Please also add "tumefaciens" for Agrobacterium and a link to the transformation protocol (or write your own transformation protocol). And provide the names of the CRISPR/Cas9 vectors. (lines 623-624)

How did you determine that there were no off-target cuttings? If you carried out PCR and sequencing of possible off-targets, then include the primer sequences in Table S6.

If possible, publish sequencing data (raw reads).

Again, make sure that all genes and species are written in italics.

Author Response

For research article

Response to Reviewer 2 Comments

1. Summary

Thanks for your comments concerning our manuscript entitled " Whole-Transcriptome Profiling and Functional Prediction of Long Non-Coding RNAs Associated with Cold Tolerance in Japonica Rice Varieties". These comments are valuable and helpful for us to revise our manuscript and guide our future research. We had carefully reviesed the paper according to these comments, and we hope this revision would be approved by you. The main changes in the manuscript and responses to these comments are listed below.

2. Questions for General Evaluation

Reviewer’s Evaluation

Response and Revisions

Does the introduction provide sufficient background and include all relevant references?

Yes/Can be improved/Must be improved/Not applicable

Are all the cited references relevant to the research?

Yes/Can be improved/Must be improved/Not applicable

Is the research design appropriate?

Yes/Can be improved/Must be improved/Not applicable

Are the methods adequately described?

Yes/Can be improved/Must be improved/Not applicable

Are the results clearly presented?

Yes/Can be improved/Must be improved/Not applicable

Are the conclusions supported by the results?

Yes/Can be improved/Must be improved/Not applicable

3. Point-by-point response to Comments and Suggestions for Authors

Comments 1: Please also add "tumefaciens" for Agrobacterium and a link to the transformation protocol (or write your own transformation protocol). And provide the names of the CRISPR/Cas9 vectors. (lines 623-624)

Response 1: Thank you very much for your sincere suggestion. We have added the Agrobacterium transformation protocol and supplemented the name of CRISPR/Cas9 vector as suggested by you. (lines 673-697, 674)

Comments 2: How did you determine that there were no off-target cuttings? If you carried out PCR and sequencing of possible off-targets, then include the primer sequences in Table S6.

If possible, publish sequencing data (raw reads).

Again, make sure that all genes and species are written in italics.

Response 2: Thank you very much for your sincere suggestion. We didn't test for off-target cuttings through sequencing. We are through biology website CRISPR - GE (http://skl.scau.edu.cn/) and NCBI grna (https://blast.ncbi.nlm.nih.gov) evaluation target specificity. (lines 670-673)

Comments 3: Again, make sure that all genes and species are written in italics.

Response 3: Thank you very much for your sincere suggestion. We have checked and corrected the species and genes in italics in accordance with your request. (lines44, 94-95, 97, 102-103, 107-108, 111, 412, 463-465, 484, 488, 490, 497-498, 504-508, 510-511, 517-518, 520, 662, 718-719)

Reviewer 3 Report

Comments and Suggestions for Authors

The authors have modified the manuscript as suggested, however, I have some of the following suggestions in respect to the current version of manuscript:

It would be better to use proline instead of an abbreviation PRO

Section 2.1.1: It would be better if authors could include the quantitative data of plant height and degree of curling of leaves in the manuscript.

Line#198 additional comma seem to be there

Line# 208-209 "The number of mRNAs not on a chromosome is more than lncRNA" seems to be confusing. 

Line#216 to 217: the suggestion provided by the authors seems to be basic. The section should be strengthen by including some statistical significance.

I think authors should include the enrichment analysis of differentially expressed mRNA and lncRNA. 

Comments on the Quality of English Language

The language seems to be fine, except for some typos.

Author Response

For research article

Response to Reviewer 3 Comments

1. Summary

Thanks for your comments concerning our manuscript entitled " Whole-Transcriptome Profiling and Functional Prediction of Long Non-Coding RNAs Associated with Cold Tolerance in Japonica Rice Varieties". These comments are valuable and helpful for us to revise our manuscript and guide our future research. We had carefully reviesed the paper according to these comments, and we hope this revision would be approved by you. The main changes in the manuscript and responses to these comments are listed below.

2. Questions for General Evaluation

Reviewer’s Evaluation

Response and Revisions

Does the introduction provide sufficient background and include all relevant references?

Yes/Can be improved/Must be improved/Not applicable

Are all the cited references relevant to the research?

Yes/Can be improved/Must be improved/Not applicable

Is the research design appropriate?

Yes/Can be improved/Must be improved/Not applicable

Are the methods adequately described?

Yes/Can be improved/Must be improved/Not applicable

Are the results clearly presented?

Yes/Can be improved/Must be improved/Not applicable

Are the conclusions supported by the results?

Yes/Can be improved/Must be improved/Not applicable

3. Point-by-point response to Comments and Suggestions for Authors

Comments 1: It would be better to use proline instead of an abbreviation PRO

Response 1: Thank you very much for your sincere suggestion. We have changed PRO to proline in the whole article according to your request. (line 65, 70, 72, 74, 146-147, 149, 358, 430, 433, 439, 557, 700)

Comments 2: Section 2.1.1: It would be better if authors could include the quantitative data of plant height and degree of curling of leaves in the manuscript.

Response 2: Thank you very much for your sincere suggestion. We have included quantitative data of plant height and degree of curling of leaves as you suggested. (lines 134-143)

Comments 3: Line#198 additional comma seem to be there.

Response 3: Thank you very much for your sincere suggestion. We have removed the excess commas as you suggested. (line 205)

Comments 4: Line# 208-209 "The number of mRNAs not on a chromosome is more than lncRNA" seems to be confusing.

Response 4: Thank you very much for your sincere suggestion. We removed the confusing sentence as you suggested. (line 215)

Comments 5: I think authors should include the enrichment analysis of differentially expressed mRNA and lncRNA.

Response 5: Thank you very much for your sincere suggestion. We added the enrichment analysis of differentially expressed mRNA and lncRNA according to your suggestion. (lines 254-260, 298-302, 586-600)

Round 3

Reviewer 3 Report

Comments and Suggestions for Authors

The Authors have modified the manuscript as suggested. I would like to recommend  the current version of manuscript.

Comments on the Quality of English Language

The language seems to be fine to me.